# The Influence of Bitumen Nature and Production Conditions on the Mechanical and Chemical Properties of Asphalt Mixtures Containing Reclaimed Asphalt Pavement

**DOI:** 10.3390/ma18153713

**Published:** 2025-08-07

**Authors:** Emiliano Prosperi, Edoardo Bocci, Giovanni Marchegiani

**Affiliations:** 1Dipartimento di Ingegneria Civile Edile e Architettura, Università Politecnica delle Marche, Via Brecce Bianche, 60131 Ancona, Italy; e.prosperi@pm.univpm.it; 2Department of Theoretical and Applied Sciences, Università degli Studi eCampus, 22060 Novedrate, Italy; giovanni.marchegiani@studenti.uniecampus.it

**Keywords:** asphalt mixture, reclaimed asphalt pavement, hot recycling, production temperature, straight-run bitumen, visbreaker bitumen, ageing

## Abstract

Several variables influence the performance of hot asphalt mixtures including reclaimed asphalt pavement (RAP). Among these, the virgin bitumen’s origin, the mix production temperature and the time the mix is kept at a high temperature between mixing and compaction play a fundamental role but are often neglected. This study aimed to quantify the negative effects associated with the improper choice of these variables. Therefore, their influence on the mechanical (indirect tensile stiffness modulus and strength, Cracking Tolerance Index) and chemical (Fourier Transform Infra-Red spectroscopy) characteristics of asphalt mixtures containing 50% RA were investigated. In particular, two rejuvenators, two types of virgin bitumen (visbreaker and straight-run), two production temperatures (140 °C and 170 °C) and three conditioning times in the oven (30 min, 90 min and 180 min) were analyzed. The results showed interesting findings that allow us to recommend selecting the virgin bitumen type carefully and to avoid excessively stressing the binder during the production of the mix.

## 1. Introduction

The growing cost of paving material and the need to consider environmental aspects has led many researchers to develop more economical and sustainable pavements [1,2,3]. The most common approach followed to address both issues is to recycle the old failing pavement. Among its many positive aspects, the recycling of pavement materials reduces the consumption of virgin resources, which in turn cuts down the cost and the energy associated with road construction and allows the saving of valuable landfill spaces [4].

Reclaimed asphalt pavement (RAP) consists of unprocessed (milled) or processed (crushed) asphalt mixtures from roadways or other resources. Two ways to reuse RAP for the construction of new road pavements are available: the cold and hot techniques [5]. In the cold technique, RAP acts as a “black” aggregate. In the hot technique, however, heating softens the old and aged bitumen within RAP, which blends with the virgin binder and causes stiffening of the new Hot Mix Asphalt (HMA) [6]. This increase in stiffness and brittleness results in a mixture more prone to thermal and fatigue cracking [7,8].

To improve the cracking resistance of HMAs with high RAP contents, one of the approaches considered is the use of rejuvenators [9]. The goal of any rejuvenator (sometimes called a recycling agent) is to return the hardened asphalt binder close to its original viscoelastic state [10]. Therefore, rejuvenators aim to decrease the viscosity and stiffness of the aged bitumen, while improving the ductility [11,12]. Several products can be used for this aim. Although there is still a lot of confusion in the literature regarding the term “rejuvenator”, different authors have recently proposed the following classifications based on the materials’ additive effect: softening agents and real rejuvenators. The softening agents, also indicated as fluidifying additives or rheological rejuvenators, allow for reductions in aged bitumen viscosity by providing oily components, while real rejuvenators help renovate the bitumen’s chemical and physical properties by disaggregating the asphaltene clusters at the intermolecular level [11,13,14]. Among the latter, a further distinction between additives derived from oil and biological additives has been introduced [15,16]. Soft virgin binder [17], waste vegetable oil [18,19,20], waste vegetable grease [21], distilled tall oil [22,23] and waste engine oil [24,25] were found to belong to the real rejuvenator category. The increasing use of RAP in new HMA has necessitated a complete understanding of how rejuvenators modify aged bitumen’s properties. This has led researchers to widely investigate this topic in recent years [26].

Recently, Fourier Transform Infra-Red spectroscopy (FTIR) has been widely used to evaluate both the aging of bitumen and the effectiveness and diffusion of a rejuvenator [27,28]. In particular, bitumen oxidation entails significant changes in the FTIR spectrum, notably the rise of carbonyl C=O (1690 cm^−1^) and sulfoxide S=O (at 1030 cm^−1^) bands. Evaluating the amount of these chemical groups in an HMA binder allows assessment of the bitumen’s aging level [29,30,31]. Moreover, even the comparison between the spectra of the pure additive and the recovered bitumen can allow the identification of the presence of a rejuvenator in an HMA [32]. Unfortunately, since the peaks corresponding to the sulfoxide and carbonyl bands are peculiar in the additive spectra, it is complicated to precisely quantify the rejuvenator content in the recovered binder [33,34]. Furthermore, different additives exhibit peculiar bands. Thus, in order to hypothetically quantify the amount of rejuvenator contained in an HMA, an FTIR test on the pure additive and an accurate correlation curve between the height/area of a particular peak and the rejuvenator content are required.

Among the several aspects that influence the performance of an HMA, the origin of the virgin binder and the temperature of the components in the mixing step play a fundamental role [35,36]. In general, HMAs are manufactured by mixing RAP, virgin bitumen, virgin aggregates and rejuvenating agents. Therefore, not only the proper rejuvenation of the RAP binder but also the performance of the virgin bitumen is essential to ensure good properties of the mixture and avoid premature pavement failure [37]. Virgin asphalt binders with the same Performance Grade (PG) could lead to mixtures with different performances because of the crude oil source and the type of distillation process [38]. In particular, a typical crude oil distillation process provides two steps. In the first step the lighter crude oil components are isolated by reaching a temperature of approximately 350 °C at atmospheric pressure. In the second step, the residue of the previous processing is brought to a temperature between 350 °C and 425 °C at a controlled pressure ranging from 1 to 10 kPa. The residue of the second process is called straight-run bitumen [39]. In the visbreaking process, the straight-run bitumen is subjected to a further thermal distillation at temperatures between 455 °C and 510 °C [40]. While this process allows refineries to reduce the amount of residue produced and recover lighter products like diesel and gas, it simultaneously penalizes the quality of the bitumen that is obtained. In fact, visbreaker (VB) bitumen is more rigid, brittle and susceptible to aging [41,42]. However, pavement technologists distinguish bitumen only by penetration or PG, without considering the distillation process from which it is derived. Given the amount of VB bitumen sold in Europe and the scientific literature on this aspect, more studies are needed to deeply investigate the influence of the oil distillation process on HMA characteristics.

During mix production at the plant, it is fundamental to manage the temperatures of each component, in particular when high RAP contents are included. Recently, Rathore and Zaumanis, evaluating the impact of the laboratory mixing procedure on the properties of HMA, highlighted that heating and mixing temperatures are critical parameters for high-RAP mixtures [43]. In particular, when RAP is added to the mix cold, its heating is achieved through the virgin aggregate, which is consequently overheated to obtain an adequate final temperature for the mix and good compactability. However, the adoption of high temperatures can cause more severe aging for both virgin and RAP bitumen, increasing mixture stiffness and brittleness [44]. Moreover, if loose HMA is kept at a high temperature for a long time (e.g., when compacting many specimens from a single batch in the laboratory) the detrimental effect of this short-term aging is further emphasized [35,44]. On the other hand, a low mixing temperature can entail poor workability and thus a high air void content. Consequently, this can increase the risk of moisture damage, raveling, rutting and cracking [45]. Therefore, it is important to balance the temperatures of both the raw materials and the mixture.

In the light of the presented scenario, it is evident that during the production of HMA including RAP, there are some variables whose paramount importance is often underestimated: the virgin bitumen distillation type, the HMA mixing temperature and time and the temperature sensitivity of the rejuvenator. This study aims to fill this gap in knowledge, quantifying the detrimental effects caused by the improper selection of these variables.

## 2. Materials and Methods

### 2.1. Objective and Experimental Program

The main goal of this research was to compare the mechanical and chemical properties of HMA with RAP when varying the type of rejuvenator, the origin of the virgin bitumen, the mix production temperature and the conditioning time of the loose mixtures in the oven (the time elapsed between mixing and compaction). In particular, two rejuvenators (coded with the letters A and B), two types of virgin bitumen (visbreaker and straight-run), two production temperatures (140 °C and 170 °C) and three conditioning times of the loose mix in the oven (30 min, 90 min and 180 min) were included in the experimental program. All the HMA mixtures contained 50% RAP. In addition, a reference mix with no RAP was investigated.

It is highlighted that no warm-mix additives, such as waxes or surfactants, have been used. The temperature of 140 °C is considered the standard temperature that should be adopted during mix production based on the selected bitumen type. The temperature of 170 °C was investigated in order to observe how the mix properties change when an excessive overheating of mix components is achieved to compensate for cold RAP addition. Moreover, the different conditioning times of the loose mixes in the oven were investigated in order to simulate different hauling times and compare the visbreaker and straight-run bitumens in terms of aging resistance.

The experimental program included indirect tensile stiffness modulus (*ITSM*) and indirect tensile strength (ITS) tests. The Cracking Tolerance (CT) Index was calculated from ITS test data in order to evaluate the ductile features of the mixtures according to ASTM D8225 [46]. The variation of the HMA properties as a function of the production temperature was assessed by introducing a so-called Temperature Index (*TI*). The details about *TI* formulation are reported in the next section of the paper. Finally, FTIR analysis was carried out on the bitumen extracted from the mixtures in order to explore the effect of bitumen type, production temperature and conditioning time in the oven at a chemical level. Figure 1 summarizes the experimental program.

### 2.2. Materials

The mixtures without RAP (00RAP) and containing 50% RAP by aggregate weight (50RAP) were designed following the Italian specifications for a binder course. The virgin aggregate (coarse fractions, sand and filler) was limestone. Two fractions of RAP with a particle size lower than 8 mm and between 8 and 16 mm, respectively, were used. The RAP binder contents, assessed through bitumen extraction using trichloroethylene, were 5.1% for the 0–8 fraction and 4.8% the 8–16 fraction by RAP weight, respectively. The aged bitumen coming from the RAP showed a penetration of 13 × 10^–1^ mm at 25 °C and a softening point of 77.1 °C. Moreover, the bitumen extraction allowed us to define the “white” gradation curve of the RAP fractions, which was considered in the mix design of the aggregate blend.

Figure 2 shows that the gradation curves of the mixtures with and without RAP were similar and laid within the reference envelope given by the Italian specifications for a binder course.

Two 50/70 virgin bitumens were used: a straight-run bitumen (SR) and one obtained as the residue from a visbreaking process (VB). Both binders are typically used in Italy to produce asphalt mixtures. Table 1 shows the physical and rheological characteristics of the two bitumens (investigated in a previous study [36]), denoting very similar properties in unaged conditions despite their different oil distillation processes.

Based on a volumetric mix design, the total binder content was fixed to 5.2% by mix weight for all the investigated HMAs. Considering a 100% reactivation of the aged binder contained in the RAP aggregates, the mixtures with 50% RAP included 2.9% virgin bitumen by mix weight. Two different commercial rejuvenators were used in this study. According to the information provided by the producers, the rejuvenators consisted of the following:A: A mix of different chemicals, containing modified polyamines and vegetal oils.B: A miscible crude tall oil derived from the processing of pine wood in the paper industry, containing fatty acids, resin acids and unsaponifiable matter.

According to the recycling agent classification procedure proposed by BRRC Dossier 21 [14], rejuvenator A is a combination of vegetal oils from agro-industry (group 4) and various specifically engineered additives (group 6), while rejuvenator B is an engineered bio-based oil (group 5). Both rejuvenator A and rejuvenator B can be classified as products with a dispersant effect (Code C). Based on a previous study [47], the additive contents were 9% and 6% by RAP binder weight for rejuvenator A and rejuvenator B, respectively. Table 2 shows the physical properties of the rejuvenators as declared by the producers.

FTIR analysis was carried out on the pure rejuvenators. The absorbance spectra in Figure 3 show that both additives have a complex chemical structure, related to the presence of many peaks with wavenumbers <1200 cm^−1^. Moreover, the two rejuvenators have unsaturated compounds (band at 3010 cm^−1^) and ester C=O groups (band at 1742 cm^−1^). In particular, it has to be remarked that the ester band at the wavenumber 1742 cm^−1^ is highly prominent for rejuvenator B. Differently, rejuvenator A has a smaller ester band but presents peaks corresponding to N-H bends (1589 cm^−1^) and N-H stretches (3400 cm^−1^), denoting the presence of amines.

The laboratory mixing procedure followed EN 12697-35. Specifically, the virgin aggregate and RAP were heated in the oven at 140 °C or 170 °C for 3 h, while the virgin binder was heated for 2 h. Two ways to add the additives to the mixtures were used: additive A was blended for 10 min with the preheated (1 h and 30 min) neat binder, using a mechanical mixer. Then, the virgin binder containing the additive was kept in the oven for 30 min before mixing with the aggregates. Differently, additive B was sprayed on the RAP at room temperature. Then, the RAP dampened with the additive was placed in the oven together with the other virgin materials to reach the target mixing temperature. The materials were mixed using a mechanical mixer: the coarse aggregates and RAP were initially dry-mixed; then, the bitumen and, in the end, the filler were added. After mixing, the loose asphalt mixtures were kept in the oven for different times (30, 90 and 180 min). At the end of each time, two cylindrical specimens with 100 mm diameter were made using a Superpave gyratory compactor, fixing the height (58 mm) and the mass (1080 g) of the specimens in order to obtain samples with an air void content of approximately 4%. The number of gyrations required by the specimens to reach the target height ranged between 50 and 100, and no significant difference between the various mixtures was identifiable. This denoted a good mix compactability independent from the virgin bitumen type, the presence of RAP and a rejuvenator, the production temperature and the time in the oven before compaction. In particular, it is remarked that for the virgin bitumen used (50/70 pen), it was not necessary to raise the mix components above a certain temperature to achieve good workability.

### 2.3. Test Methods

The indirect tensile stiffness modulus (*ITSM*) tests were performed after a 3 h conditioning of the specimens at 20 °C. The tests were carried out in control deformation using a servo-pneumatic Nottingham Asphalt Tester device according to EN 12697-26—Annex C. During the test, pulse loads with a rise time of 124 ms were applied to achieve a target horizontal deformation of 3 μm. The *ITSM* is defined as in Equation (1):(1)$$ITSM=\frac{F\cdot \left(v+0.27\right)}{\left(z+h\right)}$$
where *F* is the maximum vertical load applied, *z* is the horizontal deformation, *h* is the specimen thickness and *ν* is Poisson’s ratio, assumed to be 0.35 [48]. Four specimens for each mix were tested.

The indirect tensile strength (*ITS*) was measured at the temperature of 25 °C through an electro-mechanical device, applying a constant rate of deformation of 50 ± 2 mm/min until specimen failure occurred, according to EN 12697-23. In particular, the *ITS* was calculated through the following Equation (2):(2)$$ITS=\frac{2\cdot F}{\pi \cdot L\cdot D}$$
where *F* is the maximum vertical load, *L* is the specimen thickness and *D* is its diameter. Cracking Tolerance Index (CT-Index) was calculated according to ASTM standards [46] as an indicator of mix ductility [49]. In particular, the lower the CT-Index, the stiffer and more brittle the asphalt mixture. The CT-Index can be calculated by the following Equation (3):(3)$$\mathrm{CT-Index}=\frac{t}{62}\cdot \frac{G_{f}}{\frac{P}{l}}\cdot \left(\frac{l}{D}\right)$$
where *t* and *D* are, respectively, the mean specimen thickness and diameter; the fracture energy *G_f_* is the work of fracture (the area of the load vs. vertical displacement curve) divided by the area of the cracking face (*t ∙ D*); *l* and *P/l* are, respectively, the displacement and the slope of the load–displacement curve when the load is reduced to 75% of the peak. In particular, the ratio *P/l* can be calculated through Equation (4):(4)$$\frac{P}{l}=\left|\frac{P_{85}+P_{65}}{l_{85}+l_{65}}\right|$$

*ITS* and the CT-Index were calculated as averages of 4 specimens. In order to provide a statistical validation of the findings, the test results were examined through analysis of variance (ANOVA), assuming a significance level of 0.05 and both one-way (influence of time) and two-way (influence of temperature and type of bitumen) configurations [50,51].

With the aim to quantify the influence of the production temperature on the characteristics of HMA, changing the type of virgin bitumen, the rejuvenator and the conditioning time in the oven of the loose mixture, a Temperature Index (*TI*) was introduced. The *TI* was defined as the ratio between the difference in the quantity *X* (*ITSM*, *ITS* or *logCT-Index*) of the mixtures made at 170 °C and 140 °C divided by a reference value adopted for comparison. Specifically, the *TI* for the generic mixture “*Mix*” and the quantity “*X*” was calculated as follows:(5)$${TI}_{X,Mix}[\%]=100\cdot \frac{\left|X_{Mix,170{}^{\circ}C}-X_{Mix,\;140{}^{\circ}C}\right|}{X_{ref}}$$

The reference value *X_Ref_* was coincident with *X_Mix,140°C_* for *ITSM* and *logCT-Index*, while for *ITS* it assumed a fixed value of 1.06 MPa, corresponding to the intermediate value of the acceptability range provided by Italian specifications [52]. The global *TI* value of the mixture was calculated as the sum of the single *TI*s for each quantity (*X*).(6)$${TI}_{Mix}=\sum {TI}_{ITSM,Mix}+{TI}_{ITS,Mix}+{TI}_{\log CT-Index,Mix}$$

Based on the definitions provided by Equations (5) and (6), the lower the *TI*—the closer the performances of the mixes produced at different temperatures were—the lower the temperature sensitivity of the HMA.

Finally, FTIR analysis was carried out on bitumen extracted from different mixtures. An FTIR Spectrometer in Attenuated Total Reflectance (ATR) mode with a diamond crystal was used. The spectra were collected in the 4000–600 cm^–1^ wavenumber range with a resolution of 4 cm^–1^, and each final spectrum represented an accumulation of 16 spectra. To evaluate the effect of the time spent in the oven, the bitumen extracted from the mixtures made using the VB binder at 170 °C was analyzed. Moreover, to evaluate the influence of bitumen type and production temperature, the spectra of the mixtures kept in the oven for 30 min before compaction were compared. As the oxidation reaction entails an increase in the carbonyl functional groups (C=O) in the bitumen chemical structure [53], the Carbonyl Index *I_C=O_* was determined from the FTIR spectra in order to quantify the effect of aging. The *I_C=O_* was calculated according to the following equation [54]:(7)$$I_{C=O}=\frac{A_{1690}}{A_{1460}+A_{1375}}$$
where *A*_1690_ is the area of the spectrum around the wavenumber of 1690 cm^−1^ and corresponds to the carbonyl group, while *A*_1460_ and *A*_1375_ are the areas around the peaks centered at 1460 cm^–1^ and 1375 cm^–1^ and, respectively, correspond to the reference ethylene and methyl.

## 3. Results and Discussion

### 3.1. Mechanical Tests on the Mixtures

The graphs in Figure 4 and Figure 5 depict the mechanical parameters *ITSM*, *ITS* and CT-Index as functions of the conditioning time of the loose mixtures in the oven. As expected [34,44], the presence of 50% RAP in the mixture entailed an increase in *ITSM* and *ITS* and a decrease in CT-Index with respect to the reference mix 00RAP under the same conditions (bitumen type, production temperature, conditioning time). The addition of rejuvenator A or B allowed the mitigation of the effects of the RAP by reducing the stiffness and strength and increasing the ductility. Between the two additives, rejuvenator A was more effective, as the behavior of the mix 50RAP+A was closer to that of the mix 00RAP compared to the mix 50RAP+B. This result is consistent with a previous study involving the same additive types [47]. However, it has to be considered that the dosage of rejuvenator A (9% by RAP bitumen weight) was higher than that for rejuvenator B (6% by RAP bitumen weight). Therefore, the different efficacy of the two additives could be associated with the different dosage.

Figure 4 and Figure 5 also show that, independently from the mix type (with or without RAP/rejuvenator), bitumen origin (VB or SR) and mixing temperature (140 °C or 170 °C), *ITS* and *ITSM* increased while the CT-Index decreased when increasing the conditioning time in the oven, as observed in another study [35]. This was caused by short-term bitumen aging, which was more severe when the mixtures were conditioned in the oven for a prolonged time. From Figure 5, it can be noted that the mixtures made at 170 °C (dotted lines) achieved higher *ITS* and *ITSM* and lower CT-Index values with respect to the ones made at 140 °C (solid lines). Moreover, the slope of the trend for the mixtures made at 170 °C was higher than that for the mixtures made at 140 °C. This denotes that the higher production temperature determined more severe aging of the bitumen, and that the phenomenon was emphasized when increasing the conditioning time.

The graphs in Figure 5 allow for investigating the influence of the different virgin bitumens on mix mechanical properties. It can be immediately observed that the SR bitumen (square dots and purple lines) achieved lower *ITSM* and *ITS* and higher CT-Index values compared to the VB bitumen (circle dots and blue lines) under the same conditions. In particular, it can be noted that the slopes of the purple lines (SR) are lower than the blue ones (VB), denoting lower sensitivity to the conditioning time in the oven. This result confirmed the findings from previous research [36,42]. Moreover, the distance between the *ITSM*, *ITS* and CT-Index values measured at 170 °C and 140 °C was also lower in the case of SR bitumen. These findings are clearer when observing the graphs in Figure 5a related to the 00RAP mixtures. In these mixes, all bitumen was virgin (no RAP binder was present), and the distinct behaviors of SR and VB bitumen were accentuated. The lower influence of the production temperature and conditioning time on the mixtures with SR bitumen clearly indicates the higher resistance of the straight-run bitumen against aging phenomena and their related effects.

### 3.2. Statistical Analysis

One-way and two-way ANOVA tests were carried out on the experimental data shown in the previous section to assess their statistical significance. Table 3 reports the *p*-values obtained through one-way ANOVA test on the *ITSM*, *ITS* and CT-Index data from specimens subjected to different conditioning times. As the significance *p* was always lower than the threshold of 0.05, the variation in the mechanical properties (*ITSM*, *ITS* and CT-Index) when increasing the conditioning time in the oven, independently from the mix type (with or without RAP/rejuvenator), bitumen origin (VB or SR) and mixing temperature (140 °C or 170 °C), can be considered statistically consistent. The two-factor ANOVA test was carried out to estimate the impact of the production temperature and the bitumen type on the mechanical properties. The significance values presented in Table 4 are distinguished between those indicating the influence of the bitumen type (bitumen factor) and those representing the effect of the production temperature (T factor). Table 4 shows that the significance values *p* were always lower than 0.05, confirming the assumption that bitumen type and mixing temperature have a great influence on the mix properties (*ITSM*, *ITS* and CT-Index). Some red *p*-values (corresponding to a significance >0.05) were obtained for the 00RAP mix. Specifically, when the conditioning time was low (30 min), the mixing temperature showed a low influence on all the mechanical parameters, while the bitumen origin showed a low influence only on *ITSM*. When conditioning in the oven was extended to 90 min or 180 min, the variation in the mechanical properties when changing bitumen type and mixing temperature was statistically consistent (significance < 0.05) for the mixes with no RAP.

### 3.3. Variation in Temperature Index

The influence of the production temperature on the mix behavior was investigated by analyzing the Temperature Index (*TI*) values shown in Figure 6. First, it can be observed that all *TI* values were positive, denoting an overall lower performance (higher *ITS* and *ITSM*, lower CT-Index) for the mixtures produced at 170 °C compared to those manufactured at 140 °C, as observed in a previous study [55]. Moreover, it can be noted that, in general, the *TI* increased when increasing the conditioning time of the mix in the oven, assessing that the severity of short-term aging is directly proportional to the time during which the mix is exposed to high temperatures. Between the two types of virgin bitumen, mixtures including VB showed higher *TI* values than the mixtures including SR in most cases. This confirms that bitumen produced through a visbreaking process is more unstable and prone to oxidizing. Finally, the graph in Figure 6 shows that there is no trend of *TI* as a function of the presence of RAP or rejuvenator. In particular, the *TI* values of mix 50RAP were lower than those of mix 00RAP when VB virgin bitumen was used, and vice versa for the mixes with SR. Similarly, the *TI* of mix 50RAP+B was higher than those of mix 50RAP when VB virgin bitumen was used, while the opposite was obtained for the mixes with SR. Only mix 50RAP+A showed higher *TI* compared to mix 50RAP whether VB or SR was used. This result indicates the tendency of rejuvenator A to reduce its efficacy when increasing the mixing temperature, possibly because of evaporation (indeed, the Flash Point of rejuvenator A was close to 170 °C, as previously reported in Table 2).

### 3.4. Analysis of FTIR Spectra

The results of FTIR analysis on the binders extracted from the different mixtures are shown in Figure 7 and Figure 8. In particular, Figure 7 presents the FTIR spectra, focusing on the wavenumbers between 1300 and 1900 cm^–1^, of the bitumens from the mixes including VB, produced at 170 °C and conditioned for 180 min. It can be noted that the bitumen extracted from the 00RAP mix showed a clear presence of carbonyl groups (band at 1690 cm^–1^), denoting that the high mixing temperature (170 °C) and prolonged time in the oven (180 min) determined significant oxidation of the VB virgin binder. However, the peak of this bitumen was still shorter than that for the binders extracted from the HMA mixtures including 50% RAP. In particular, the short-term-aged virgin bitumen and the severely aged RAP bitumen (the natural long-term aging suffered on site and the short-term aging undergone in the lab) fully blended during the extraction process, resulting in the marked presence of oxidation products (C=O groups) in the spectra. The binders extracted from the 50RAP, 50RAP+A and 50RAP+B mixes showed a comparable height of the peaks at 1690 cm^–1^, denoting that the rejuvenators had no effect on the chemical bonds generated through oxidation. However, the presence of the rejuvenators could be noticed from the bands at 1742 cm^–1^, which was evident in the spectra of 50RAP+A bitumen and, even more so, of 50RAP+B bitumen. In fact, the spectra of the rejuvenators (Figure 3) showed a characteristic ester band (higher for rejuvenator B) at the wavenumber of 1742 cm^–1^, which was reflected on the spectra of the extracted binders. This highlights the possibility of exploiting FTIR to assess the presence of rejuvenators in HMA and even estimate the dosage.

Figure 8 shows the Carbonyl Index *I_C=O_* of the extracted binders as a function of the conditioning time (a) and of the virgin bitumen type and production temperature (b). It can be immediately observed that the *I_C=O_* of the 00RAP bitumen was significantly lower than that for the binders of the HMA mixes, independently from the conditioning time, bitumen type and mixing temperature. The VB_170 mixes including 50% RAP showed similar values of the *I_C=O_* either without a rejuvenator, with rejuvenator A, or with rejuvenator B. This indicates that the two rejuvenators did not operate on the oxidation products at a chemical level. The *I_C=O_* increased when increasing the conditioning time that the mixtures spent in the oven before compaction. In the case of the 00RAP mix, this was related to the increasing severity of the short-term aging of the virgin VB bitumen, while for the 50RAP mixtures both the virgin and RAP bitumen experienced a more detrimental aging. For the VB virgin bitumen, a mixing temperature of 140 °C entailed a reduction in the *I_C=O_* compared to 170 °C, confirming what was observed from the mechanical tests on the HMA specimens, i.e., that the higher the mix production temperature, the more marked the effects of aging. However, the FTIR analysis of the binders extracted from the mixes including SR virgin bitumen showed different results compared to the mechanical tests on the mixtures. Specifically, for the reference HMA 00RAP with SR virgin bitumen, the *I_C=O_* values were lower than those for the analogous mix with VB bitumen, confirming that the straight-run bitumen has a lower sensitivity to aging. On the contrary, for the 50RAP mixes with/without a rejuvenator, the *I_C=O_* was higher when the SR virgin bitumen was used. This result can perhaps be justified by a disturbance during FTIR specimen sampling. Further analysis will be carried out to clarify the reason for this finding.

## 4. Conclusions

In this paper, some variables, whose importance is sometimes underestimated in the RAP hot recycling process, were investigated, specifically the bitumen origin, the mix production temperature and the time the mix is kept at a high temperature between mixing and compaction. As these variables may significantly influence the final mix behavior by affecting the binder performance and the effectiveness of the rejuvenators, this study aimed to quantify their impact. The results of the mechanical and chemical tests carried out in the laboratory showed the following:The use of SR virgin bitumen allowed the achievement of better mechanical mix behavior (lower *ITS* and *ITSM*, higher CT-Index) compared to VB virgin bitumen. This result can likely be traced back to the higher stability of the SR binder, related to its different petroleum refining process. However, the FTIR analysis of the bitumen extracted from the mixtures gave unreliable results (the *I_C=O_* was lower in the case of the SR binder for the mix without RAP, and vice versa for the mixtures with 50% RAP).When increasing the HMA production temperature from 140 °C to 170 °C, there was an increase in *ITSM*, *ITS* and *I_C=O_* and a decrease in CT-Index. This penalization of the binder/mix properties was related to the more severe short-term aging that both the virgin and RAP bitumen experienced at the higher production temperature. Therefore, in HMA plant manufacturing, the adoption of the lowest temperature that achieves good mix compactability is suggested in order to preserve the binder’s original properties.The longer the conditioning time in the oven at high temperature between mixing and compaction, the worse the behavior of the mixtures. When virgin and RAP bitumen were kept at high temperature for a long time, aging phenomena, particularly oxidation and volatile evaporation, were emphasized, resulting in a higher stiffness and brittleness (lower *ITS*, *ITSM* and *I_C=O_*, lower CT-Index). This allows us to recommend avoiding long hauling distances in full-scale pavement construction and limiting mix reheating times in the laboratory during HMA quality controls.The Temperature Index (*TI*) proved to be a useful tool to evaluate the detrimental effect of high production temperatures and prolonged storage times of the mix at high temperature. Moreover, it can allow the estimation of whether a rejuvenating agent can suffer from a reduction in efficacy when the hot recycling of RAP is carried out at high temperatures.

The present research showed interesting findings that provide warnings for HMA producers, administrations and laboratories to carefully select the virgin bitumen type and avoid excessively stressing the binder. However, these recommendations should be validated by further analyses on a wider set of materials (bitumen types, rejuvenators) and supplementary laboratory tests (fatigue and rutting tests).

## Figures and Tables

**Figure 1 materials-18-03713-f001:**
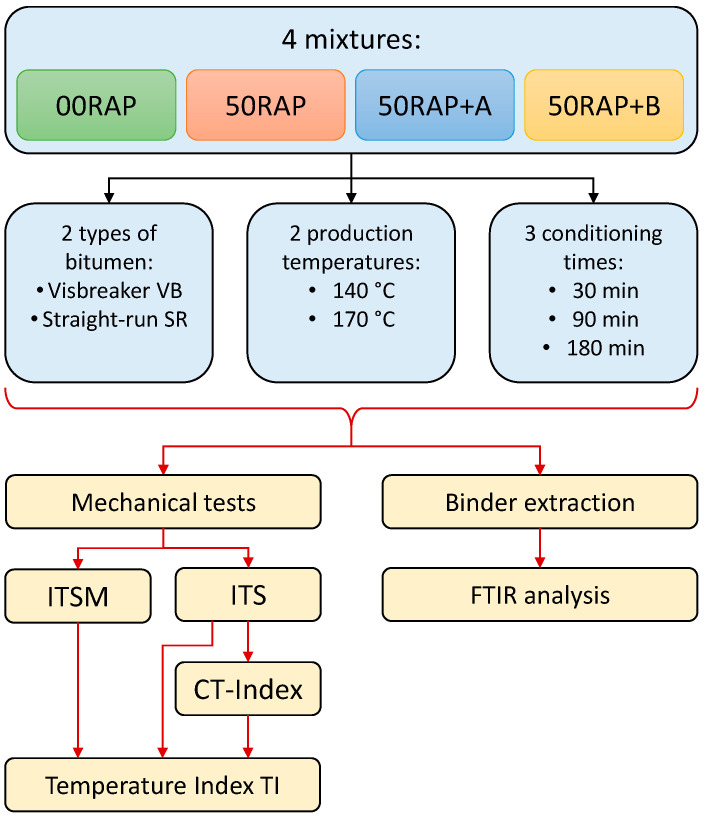
Experimental program.

**Figure 2 materials-18-03713-f002:**
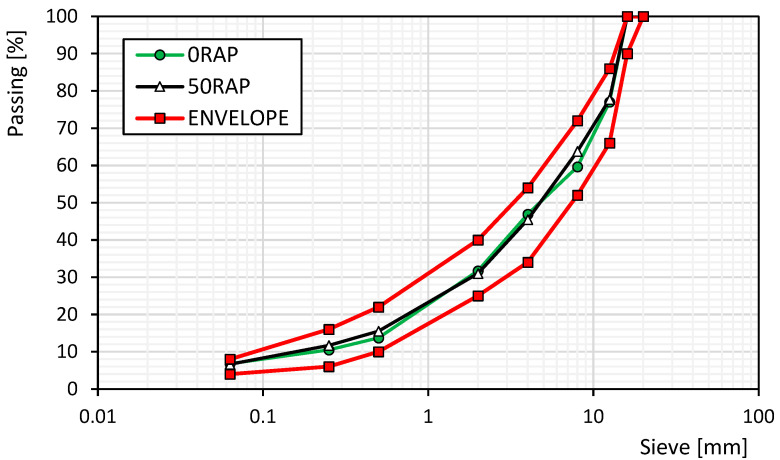
Aggregate gradation of mixtures with/without RAP.

**Figure 3 materials-18-03713-f003:**
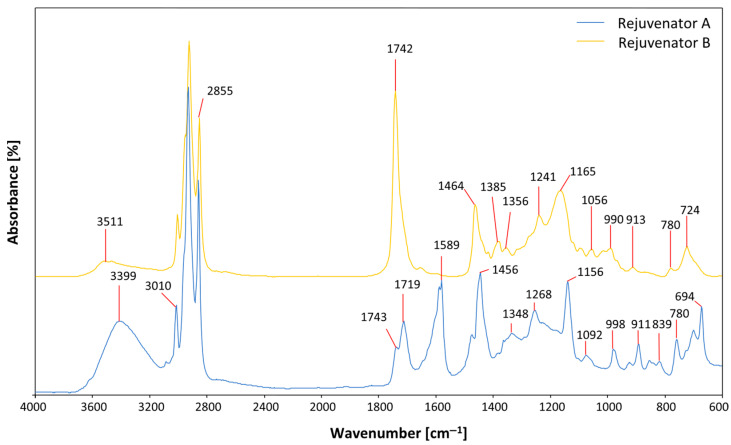
FTIR spectra of the two rejuvenators.

**Figure 4 materials-18-03713-f004:**
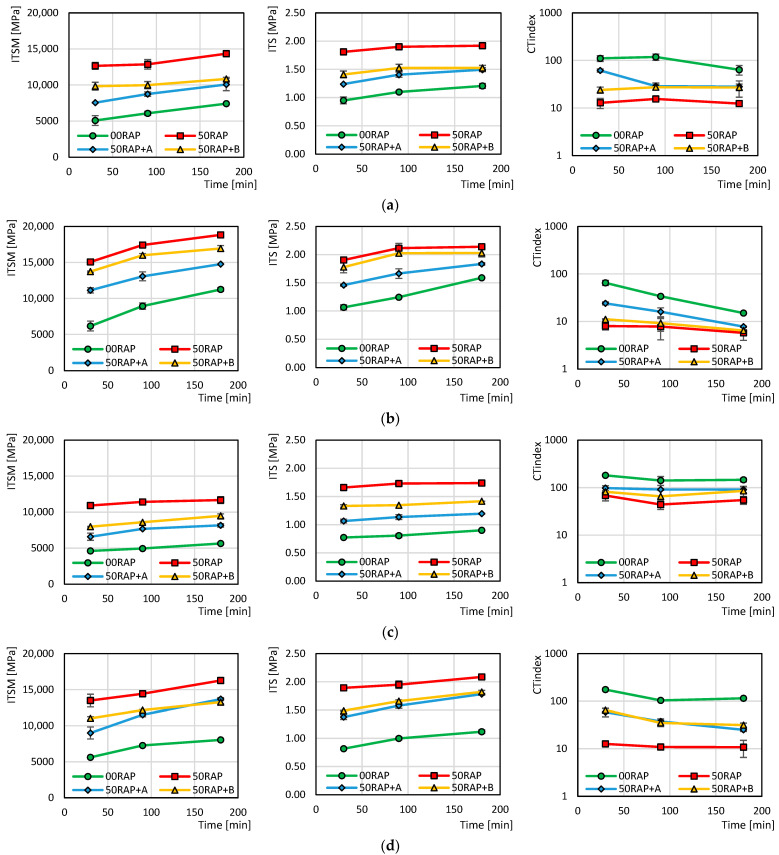
The effect of the conditioning time on the different mixtures: (**a**) VB, 140 °C; (**b**) VB, 170 °C; (**c**) SR, 140 °C; (**d**) SR, 170 °C.

**Figure 5 materials-18-03713-f005:**
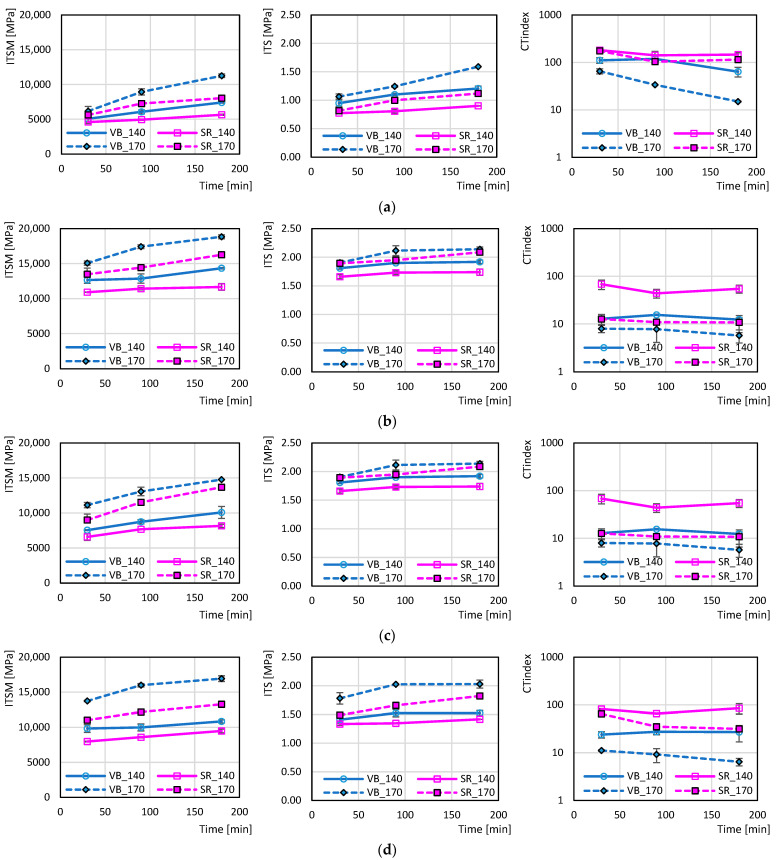
The effect of the conditioning time as a function of virgin bitumen type and conditioning temperature: (**a**) 00RAP; (**b**) 50RAP; (**c**) 50RAP+A; (**d**) 50RAP+B.

**Figure 6 materials-18-03713-f006:**
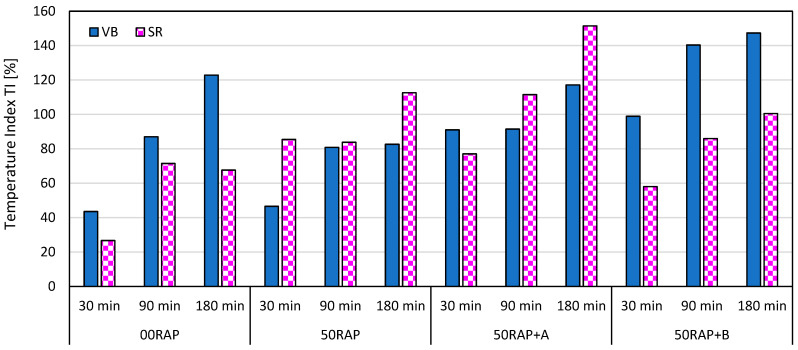
Temperature Index.

**Figure 7 materials-18-03713-f007:**
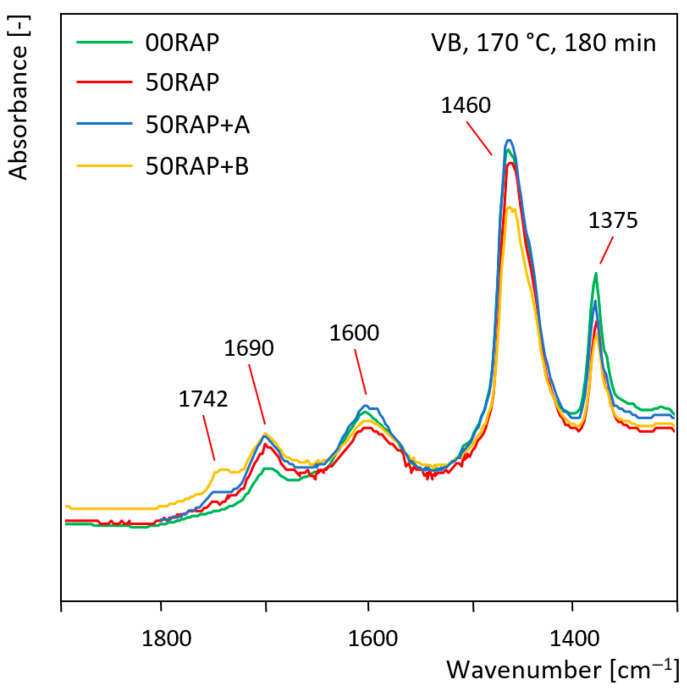
FTIR spectra of the bitumen extracted from the mixes including VB, produced at 170 °C and conditioned for 180 min.

**Figure 8 materials-18-03713-f008:**
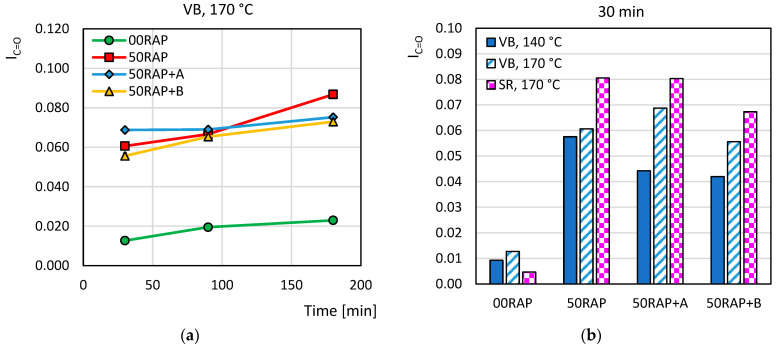
Values of *I_C=O_* as functions of (**a**) conditioning time and (**b**) bitumen type and mix production temperature.

**Table 1 materials-18-03713-t001:** Physical and rheological properties of virgin bitumens.

Property	Unit	Norm	VB	SR
Penetration at T = 25 °C	10^−1^ mm	EN 1426	62	63
Ring & ball softening point	°C	EN 1427	50	49
Temperature G */sinδ > 1 kPa	°C	EN 14770	64	64
Glover–Rowe parameter at T = 15 °C, ω = 0.005 rad/s	Pa	-	189	75

**Table 2 materials-18-03713-t002:** Physical properties of rejuvenators.

ID	Type	Density @ T = 20 °C [g/cm^3^]	Flash Point [°C]	Kinematic Viscosity @ T = 25 °C [mPa∙s]
*A*	rejuvenator	0.80	>150	45
*B*	rejuvenator	0.93	>295	98

**Table 3 materials-18-03713-t003:** *p*-values from the one-way ANOVA test evaluating the statistical significance of the influence of the loose mix conditioning time.

	*ITSM*	*ITS*	CT-Index
**30 min vs. 90 min**	0.00001	0.00001	0.0003
**90 min vs. 180 min**	0.0005	0.001	0.0308

**Table 4 materials-18-03713-t004:** *p*-values from the two-way ANOVA test evaluating the statistical significance of the influence of the mixing temperature and the bitumen type.

	*ITSM*		*ITS*		CT-Index	
	Bitumen Factor	T Factor	Bitumen Factor	T Factor	Bitumen Factor	T Factor
**00RAP-30 min**	0.3474	0.0947	0.0049	0.1041	0.0010	0.0688
**00RAP-90 min**	0.0074	0.0008	0.0003	0.0020	0.0583	0.0251
**00RAP-180 min**	0.0001	0.0001	0.0001	0.0004	0.0008	0.0157
**50RAP-30 min**	0.0331	0.0087	0.0479	0.0046	0.0196	0.0193
**50RAP-90 min**	0.0048	0.0006	0.0406	0.0182	0.0341	0.0150
**50RAP-180 min**	0.0008	0.0001	0.0344	0.0016	0.0141	0.0113
**50RAP+A-30 min**	0.0405	0.0045	0.0107	0.0007	0.0156	0.0123
**50RAP+A-90 min**	0.0240	0.0004	0.0363	0.0036	0.0006	0.0015
**50RAP+A-180 min**	0.0300	0.0004	0.0013	0.0001	0.0045	0.0035
**50RAP+B-30 min**	0.0017	0.0003	0.0361	0.0115	0.0001	0.0106
**50RAP+B-90 min**	0.0008	0.0001	0.0014	0.0003	0.0009	0.0026
**50RAP+B-180 min**	0.0011	0.0001	0.0221	0.0005	0.0240	0.0345

## Data Availability

The original contributions presented in this study are included in the article. Further inquiries can be directed to the corresponding author.

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
