# Peer review of "The Influence of Bitumen Nature and Production Conditions on the Mechanical and Chemical Properties of Asphalt Mixtures Containing Reclaimed Asphalt Pavement"

_materials, 2025, doi:10.3390/ma18153713_

Round 1

Reviewer 1 Report

Comments and Suggestions for Authors

In this manuscript, it investigates the effect of two rejuvenators, two types of virgin bitumen (visbreaker and straight-run), two production temperatures (140 °C and 170 °C) and three conditioning times in the oven (30 min, 90 min and 180 min) on the mechanical (indirect tensile stiffness modulus and strength, cracking tolerance index) and chemical (Fourier Transform Infra-Red spectroscopy) characteristics of the bituminous mixtures containing 50% of RA. The results showed interesting findings that allow recommending to select the virgin bitumen type carefully and to avoid excessively stressing the binder during the production of the mix. However, the full manuscript still has a few problems which need to be revised.

  1. It is suggested that the author make key revisions to the English in the article, as there are numerous expression and grammatical errors in it. This will make it very difficult for readers to understand what the author intends to convey.
  2. This paper is relevant to your study and should be considered for citation in the introduction. https://doi.org/10.1016/j.conbuildmat.2025.141232
  3. It is suggested that the author unify the professional terms in the text. For example, "asphalt mixtures" and "bituminous mixtures" should be unified. The author should carefully revise the content of the entire text.
  4. The singular and plural forms of professional terms should also be consistent throughout the text, such as "asphalt mixtures" and "asphalt mixture".
  5. It is suggested that the author keep the form of Equation 1 consistent with other equations.
  6. When the authors analyze the experimental results, they can accordingly cite the research achievements of predecessors and their own conclusions for verification. This can enhance the reliability and scientific nature of the research results of the article.
  7. Before analyzing the results in Section 3.3, the author should clearly explain the physical significance of the temperature index, such as what a larger temperature index means for the mixture. Just like the explanation of CT-Index (“the lower the CT-Index, the stiffer and the more brittle the bituminous mixture is”) in the previous text.
  8. It is suggested that the author write the conclusion section in the order of "3. Results and discussion", which will make the logical thinking of the article clearer.

Reviewer 2 Report

Comments and Suggestions for Authors

The submitted study addresses an actual topic – the application of reclaimed asphalt in the production of asphalt mixtures. The study is properly conducted; however, I cannot say that it presents groundbreaking or surprising findings. I consider the main shortcoming of the article to be the very small number of tested mixtures and rejuvenators, which does not allow the results to be generalized. Nevertheless, the results at least indicate how the mechanical and chemical properties of HMA may change depending on the asphalt origin, type of rejuvenator, and production conditions. I summarize my comments to be addressed below:

  • Throughout the text, the abbreviations “RA” and “RAP” are used. I recommend standardizing them.
  • As the type of rejuvenator is one of the main variables in the HMA, I would expect a more detailed description of rejuvenator A. Currently, only a brief specification is provided in Section 2.2 Materials.
  • Table 1 summarizes the properties of the virgin bitumen. Are these values provided by the manufacturers, or were they investigated by the authors? Please clarify.
  • How many samples were subjected to the ITSM and ITS tests? Please specify.
  • What was the rate of control deformation during the ITSM test?

Reviewer 3 Report

Comments and Suggestions for Authors

Paper title:  Influence of the bitumen nature and the production conditions on the mechanical and chemical properties of asphalt mixtures containing RAP.

The paper's topic is interesting, but it has not been structured properly. The novelty of the article is missing. The following are required to be considered by the authors before the paper is accepted.

  • In the title, avoid the use of abbreviations (RAP)
  • In the introduction section, you need to clearly state the gap your research is trying to fill, as it is missing.
  • During the whole paper, you are saying Reclaimed asphalt (RA) and then provide the abbreviation of RAP, which you have not introduced.
  • The aim of the research should be clearly stated in the abstract

  • The current conclusions need to be amended as it is not presented properly. Therefore, authors are first required to state the main aim of the current research at the conclusion and then list the main findings.
  • Try to include some limitations of the current study and recommendations for future work
  • Have you considered the effect of your materials on the service life ?
  • It is also important to include some figures regarding the environmental performance and, where possible, cost analysis of your work.

Round 2

Reviewer 1 Report

Comments and Suggestions for Authors

accept

Reviewer 3 Report

Comments and Suggestions for Authors

The authors have addressed my comments and I am happy with the current version of the paper.